# Aurora B Kinase Inhibition by AZD1152 Concomitant with Tumor Treating Fields Is Effective in the Treatment of Cultures from Primary and Recurrent Glioblastomas

**DOI:** 10.3390/ijms24055016

**Published:** 2023-03-06

**Authors:** Dietmar Krex, Paula Bartmann, Doris Lachmann, Alexander Hagstotz, Willi Jugel, Rosa S. Schneiderman, Karnit Gotlib, Yaara Porat, Katja Robel, Achim Temme, Moshe Giladi, Susanne Michen

**Affiliations:** 1Department of Neurosurgery, University Hospital Carl Gustav Carus, Technical University Dresden, 01307 Dresden, Germany; 2German Cancer Consortium (DKTK), 01307 Dresden, Germany; 3National Center for Tumor Diseases (NCT/UCC), 01307 Dresden, Germany; 4Section Experimental Neurosurgery/Tumor Immunology, Department of Neurosurgery, University Hospital Carl Gustav Carus, Technical University Dresden, 01307 Dresden, Germany; 5Novocure® Ltd., 5 Nahum Het Street, Haifa 31905, Israel; 6German Cancer Research Center (DKFZ), 69120 Heidelberg, Germany

**Keywords:** glioblastoma, TTFields, Aurora B kinase, AZD1152, primary cultures

## Abstract

Tumor Treating Fields (TTFields) were incorporated into the treatment of glioblastoma, the most malignant brain tumor, after showing an effect on progression-free and overall survival in a phase III clinical trial. The combination of TTFields and an antimitotic drug might further improve this approach. Here, we tested the combination of TTFields with AZD1152, an Aurora B kinase inhibitor, in primary cultures of newly diagnosed (ndGBM) and recurrent glioblastoma (rGBM). AZD1152 concentration was titrated for each cell line and 5–30 nM were used alone or in addition to TTFields (1.6 V/cm RMS; 200 kHz) applied for 72 h using the inovitro™ system. Cell morphological changes were visualized by conventional and confocal laser microscopy. The cytotoxic effects were determined by cell viability assays. Primary cultures of ndGBM and rGBM varied in p53 mutational status; ploidy; EGFR expression and MGMT-promoter methylation status. Nevertheless; in all primary cultures; a significant cytotoxic effect was found following TTFields treatment alone and in all but one, a significant effect after treatment with AZD1152 alone was also observed. Moreover, in all primary cultures the combined treatment had the most pronounced cytotoxic effect in parallel with morphological changes. The combined treatment of TTFields and AZD1152 led to a significant reduction in the number of ndGBM and rGBM cells compared to each treatment alone. Further evaluation of this approach, which has to be considered as a proof of concept, is warranted, before entering into early clinical trials.

## 1. Introduction

Glioblastoma (GBM) is the most devastating primary malignancy of the central nervous system in adults. Current standard treatment consists of maximal and safe surgical resection followed by loco-regional radiotherapy (RT) with concomitant daily temozolomide (TMZ) chemotherapy, and then maintenance treatment with TMZ for 6 to 12 months [1]. Tumor Treating Fields (TTFields) are an anticancer modality that disrupt critical processes in cancer cells, including cell division, by delivering low-intensity, intermediate-frequency (200 kHz for GBM) alternating electric fields [2,3,4]. In a phase III clinical trial (EF-14, ClinicalTrials.gov identifier NCT00916409), the efficacy of TTFields used with TMZ maintenance treatment after maximal safe resection and chemo-radiation therapy for patients with newly diagnosed GBM (ndGBM) was reported. Median overall survival (OS) from randomization in the intent-to-treat population was 20.9 months versus 16.0 months for the TTFields-TMZ group and the TMZ–alone group, respectively, with a hazard ratio of 0.63 (95% CI, 0.53–0.76; *p* < 0.001). The significant improvement in OS was seen across all patient subgroups regardless of age, extent of resection, performance status, gender, geography or promoter methylation status of O^6^-methylguanine-DNA methyltransferase (*MGMT*) gene encoding a ubiquitous nuclear enzyme involved in the repair of alkylated DNA [5]. In contrast, the phase III clinical trial EF-11 (ClinicalTrials.gov identifier NCT00379470) investigated the effect of TTFields application as the sole therapy for recurrent GBM (rGBM) compared to the physician’s choice of chemotherapy. Although a significant prolongation of overall survival was not observed, non-inferiority of the treatment-arm was demonstrated [6]. Nevertheless, the benefit of TTFields treatment outweighed the benefit of chemotherapy, due to a comparable survival and improved quality of life of the patients [7].

The anti-tumor effect of TTFields is not yet fully understood. One model is based on the principle that TTFields exert directional forces on polar microtubules and interfere with the assembly of the normal mitotic spindle. Such interference with microtubule dynamics results in abnormal spindle formation and subsequent mitotic arrest or delay. As a result, cells die while in mitotic arrest or progress to cell division [4]. This can lead to the formation of either normal or abnormal aneuploid progeny. The formation of tetraploid cells can occur due to mitotic exit through slippage or as a result of improper cell division. Abnormal daughter cells either die in the subsequent interphase, undergo a permanent arrest, or proliferate through additional rounds of mitosis where they will be subjected to further TTFields assault [2]. Another model is based on the effects of TTFields on membrane potential and ion channels. Li et al. recently showed in a theoretical study that the effect of TTFields on thermal energy, and thus elevated Brownian motion, is larger than the effect on tubulin dimer orientation. In addition, they calculated that the dielectric forces also do not have a net-effect as commonly thought, because these forces are counteracted by Stokes forces. Alternatively, they suggest that cell membrane potential is elevated over 10% of the resting cell state and that this effects ion channels and membrane pumps, resulting in an influx of Ca^2+^ into the cell, which in turn affects microtubule polymerization [8].

A promising approach to enhance the efficiency of TTFields is the use of drugs, which synergistically act together with TTFields and prolong metaphase-anaphase transition and telophase. In particular, an extended period in metaphase and during cytokinesis most likely increases DNA damage and subsequent events, leading to mitotic catastrophe and apoptosis of tumor cells. In this regard, inhibitors or drugs interfering with components of the chromosomal passenger complex (CPC), in particular affecting Aurora B kinase [9], are prime candidates for combinatorial use with TTFields. The CPC is of central importance for mitosis as a regulator of chromosome division and cytokinesis. In addition to the enzymatic component Aurora B kinase, it consists of the three regulatory and targeting components: Survivin, Borealin, and inner centromeric protein (INCENP). Aurora B kinase activity is critically involved in correcting syntelic microtubule-kinetochore connections and therefore guarantees bi-orientation of sister chromatids to opposing spindle poles before the onset of anaphase [10,11]. Therefore, inhibition of Aurora B kinase leads to defective mitosis, polyploidy, and finally to mitotic catastrophe [12]. AZD1152-HQPA (barasertib) is a highly selective Aurora B kinase inhibitor [13] with efficacy on a wide variety of tumor entities, including acute myeloid leukemia, advanced solid tumors, and diffuse large B-cell lymphoma, which has already been investigated and validated in phase I and II studies [14,15,16]. Furthermore, in vitro and in vivo studies showed that AZD1152 caused polyploidy and non-apoptotic cell death in glioma cell lines regardless of their *p53* status [17,18,19]. This is beneficial because the *p53* status of GBM patients is closely related to disease progression and survival during chemoradiotherapy [20,21]. Although, there had been evidence that the *p53*/*p73* status might be important for the type of cellular response to selective Aurora B inhibition, we have shown that together with its essential role in the execution of cytokinesis, Aurora B, in cooperation with its partners of the CPC, safeguards segregation and chromosomal integrity independently of *p53* mutational status and therefore is critical for the survival of cells [19].

In the present study, we investigate whether the combination of TTFields with the chemical Aurora B kinase inhibitor AZD1152 enhances the anti-tumor effect on glioblastoma cells by additionally inhibiting cytokinesis. To evaluate the efficiency of this combined treatment, both established long-term glioblastoma cell lines U87-MG and p53-deficient U87-MG^shp53^, as well as primary cultures of ndGBM and rGBM, are used.

## 2. Results

### 2.1. Treatment of U87-MG and U87-MG^shp53^ with AZD1152 and TTFields Increases Cell Death

To examine whether enhanced anti-tumor effects occur independently of p53 when AZD1152 and TTFields are combined, we initially used the *p53* wild-type, long-term-established glioma cell line U87-MG and its stable *p53*-deficient counterpart U87-MG^shp53^. U87-MG proved to be highly sensitive to treatment with AZD1152 alone with a half-maximal inhibitory concentration (IC_50_) around 25 nM (Figure 1A). Response to TTFields application alone (200 kHz, 1.6 V/cm RMS) for 72 h led to a reduction in cell number of 49.0% (±2.8%), similar to previous reports. The *p53*-deficient U87-MG^shp53^ cell line was less sensitive to treatment with AZD1152 than the *p53* wild-type U87-MG cell line, with an IC_50_ around 50 nM. Similarly, TTFields application alone resulted in a slightly lower cell number reduction of 40.5% (±10.6%) in U87-MG^shp53^ compared to *p53* wild-type U87-MG. The combined treatment of TTFields and AZD1152 led to a reduction in the number of U87-MG and U87-MG^shp53^ cells relative to treatment with AZD1152 alone (Figure 1A, and Appendix A). Although there is a difference between U87 and U87-MG^shp53^ cell lines we performed a test of significance but could not substantiate a statistically significant difference, either for the single treatment with AZD1152 or in combination with TTFields.

Furthermore, cell morphological changes were observed with increasing AZD1152 concentration in both cell lines. These were intensified when AZD1152 and TTFields were combined (Figure 1B). Microscopy images of U87-MG and U87-MG^shp53^ cells stained with crystal violet after treatment revealed a slight increase in the number of multinuclear cells following TTFields application (Figure 1B, first row). A higher prevalence of multinuclear cells (marked by black arrows) was observed in U87-MG and U87-MG^shp53^ cells exposed to TTFields and low concentration of AZD1152 (25 nM) compared to cells treated with AZD1152 (25 nM) alone (Figure 1B, second row). Cells treated with TTFields and higher doses of AZD1152 (50–100 nM) demonstrated increased rates of pyknosis (marked by blue arrows) (Figure 1B, third and fourth rows; enlarged views are provided in Appendix A).

### 2.2. Established Primary Glioblastoma Cultures Show Different Sensitivity to AZD1152

To verify the enhanced anti-tumor effects of concomitant treatment with AZD1152 and TTFields observed in U87-MG and U87-MG^shp53^ in a more clinically relevant model, primary glioblastoma cultures were used. For this purpose, primary ndGBM and rGBM cell cultures were established from fresh tumor tissue from patients. As tumor tissue from patients with rGBM was pretreated with alkylating chemotherapy (temozolomide) and radiotherapy, the cytotoxic effect of additional therapy with Aurora B kinase inhibition and TTFields on these primary cultures might be different from those of ndGBMs. Therefore, we not only established primary cultures from ndGBM patients but from rGBM patients as well and treated them identically. Tumor tissues were thoroughly analyzed at the Department of Neuropathology, University Hospital Carl Gustav Carus, TU Dresden according to the current World Health Organization (WHO) classification system at the day of surgery, and reconfirmed according to current criteria of the version from 2021. The Ki-67 proliferation index varied between 5% and 60%. Furthermore, the glial origin of the tumors was confirmed by identification of glial fibrillary acid protein (GFAP) by immunohistochemistry. Moreover, all tumors had no mutation of *IDH1* R132H locus. In addition, both of the ndGBM and one of the rGBM tumors (HT16360-1) are *MGMT* promoter methylated (Table 1A). The established primary glioblastoma cultures were further characterized and verified before treatment (Table 1B). All primary glioblastoma cultures showed 91.2–99.3% expression of the cancer cell marker CD44, which is overexpressed in GBM [22], and 6.8–61.6% expression of GFAP, which is expressed by astrocytes and in astrocytic brain tumors [23]. Furthermore, ndGBM culture HT12347 and rGBM culture HT16360-1 overexpressed EGFR (78.4% and 73.2%), which occurs in approximately 60% of glioblastoma patients [24]. The ndGBM culture HT18584 and rGBM culture HT16360-1 showed an increased expression of ErbB2 (95.9% and 76.3%), which like EGFR belongs to the ErbB family of receptor tyrosine kinases. In addition, *p53* status was determined, with ndGBM cultures found to be *p53* mutant and rGBM cultures found to be *p53* wild-type.

Initially, the response of primary glioblastoma cultures to treatment with AZD1152 alone was investigated. Cells showed variable sensitivity to AZD1152 treatment regardless of their *p53* status or whether they were established from ndGBM or rGBM samples. (Figure 2A). While HT18584 (ndGBM, *p53*mut) and HT18816 (rGBM, *p53*wt) cells exhibited high sensitivity with IC_50_ around 14 nM and 18 nM, HT16360-1 (rGBM, *p53*wt) and HT12347 (ndGBM, *p53*mut), cells were less sensitive with IC_50_ around 50 nM and 75 nM. The IC_50_ of HT18328-3 (rGBM, *p53*wt) was in the middle range of 30 nM. Moreover, increasing AZD1152 concentrations led to only minor changes or a slight decrease in p53 in *p53* mutant ndGBM cultures HT12347 (20 nM: 1.1×, 50 nM: 0.9×) and HT18584 (20 nM 0.6×, 50 nM: 0.8×) (Figure 2B). However, *p53* wild-type rGBM cultures showed either increased accumulation (HT16360-1, 30 nM: 2.4×, 100 nM: 3.2×) or no p53 expression (HT18816 and HT18328-3) at all.

### 2.3. AZD1152 Increases TTFields-Induced Cytotoxic Effects in ndGBM and rGBM Primary Cultures

To investigate enhanced anti-tumor effects of combined treatment with AZD1152 and TTFields of primary glioblastoma cultures, cell viability assays were performed (Figure 3). Thereby, primary glioblastoma cultures and U87-MG responded similarly to TTFields treatment alone (200 kHz) with a reduction in cell number to a median of 49.6–58.8%. An exception was HT18584 (ndGBM, *p53*mut), which was much more resistant with a median reduction in cell number to only 79.2% (47.1–94.3%). Furthermore, HT18328-3 (rGBM, *p53*wt) exhibited high variability in cell reduction from 14.6% to 88.3% (median 56.7%). Therefore, for these two primary glioblastoma cultures, AZD1152 concentrations were chosen that reduced cell count to a median of 50% living cells (HT18584 (ndGBM, *p53*mut): 49.8% with 20 nM AZD1152; HT18328-3 (rGBM, *p53*wt): 52.5% with 15 nM AZD1152). For the other primary glioblastoma cultures and U87-MG, AZD1152 concentrations were used that resulted in a reduction in cell number to a median of 75% (68.3–80.6%) of living cells. These low effective doses of AZD1152 were selected with consideration for potential clinical translation to reduce the risk of toxicity, as AZD1152 was associated with frequent adverse events, including myelotoxicity, in clinical studies [14,15]. Moreover, it was expected that in combination with TTFields even low effective doses of AZD1152 lead to increased cytotoxic effects. Compared to TTFields treatment alone, combined treatment of AZD1152 and TTFields resulted in significantly enhanced anti-tumor effects with a median reduction in cell number in U87-MG cells of 57.8% to 40.4% (Mann–Whitney U test, *p* < 0.01) as well as in all primary glioblastoma cultures (59.6–79.2% to 26.9–44.1%, Mann–Whitney U test) regardless of their p53 status or whether they were established from ndGBM or rGBM tissue samples, although further studies are needed to elucidate the exact involvement of the p53 pathway and the effect of the well-known heterogeneous molecular composition of glioblastoma.

### 2.4. AZD1152 plus TTFields Reinforce Morphological Changes in ndGBM and rGBM Primary Cultures

In addition to the enhanced anti-tumor effects of concomitant treatment with TTFields and AZD1152 relative to each treatment alone, increased cell morphological changes were also observed in confocal laser-scanning microscopic images (Figure 4). HT18584 (ndGBM, *p53*mut), HT16360-1 (rGBM, *p53*wt) and HT18816 (rGBM, *p53*wt) exhibited slight increases in the number of multinuclear cells following AZD1152 treatment (Figure 4, second column). This was also observed in HT12347 (ndGBM, *p53*mut), HT18584 (ndGBM, *p53*mut) and HT18816 (rGBM, *p53*wt) following TTFields application (Figure 4, third column). A higher prevalence of multinuclear cells was detected in all primary glioblastoma cultures exposed to AZD1152 and TTFields compared to cells treated with TTFields or AZD1152 alone (Figure 4; quantification shown in Appendix A and enlargements for AZD1152 only and TTFields only treatment, respectively, in Appendix A). Thereby, HT18584 (ndGBM, *p53*mut), HT16360-1 (rGBM, *p53*wt) and HT18328-3 (rGBM, *p53*wt) showed particularly large cell nuclei. Concomitant treatment further resulted in an increase in cell size, especially in HT12347 (ndGBM, *p53*mut), HT16360-1 (rGBM, *p53*wt) and HT18328-3 (rGBM, *p53*wt).

### 2.5. AZD1152 plus TTFields Increase Polyploidy in U87-MG, ndGBMs and Partially in rGBMs

Since treatment with TTFields and the Aurora B kinase inhibitor AZD1152 leads to impaired mitosis and polyploidy, the DNA content was further examined using propidium iodide staining of treated U87-MG and primary glioblastoma cell cultures (Figure 5). In U87-MG and the ndGBM cell cultures HT12347 (ndGBM, *p53*mut) and HT18584 (ndGBM, *p53*mut), concomitant treatment of AZD1152 and TTFields significantly increased the percentage of cells with DNA content of 4n and >4n in comparison to control or TTFields treatment alone, while the percentage of cells with 2n DNA content significantly decreased. Similar effects were observed with AZD1152 treatment alone compared to the control. In contrast, the rGBM cell cultures showed some characteristics: HT16360-1 (rGBM, *p53*wt) and HT18328-3 (rGBM, *p53*wt) had a very low proportion of cells with a DNA content of 2n in the untreated state (HT16360-1: 6.4% ± 1.7%; HT18328-3: 10.1% ± 2.2%), indicating that these are predominantly tetraploid cell cultures. In HT18328-3 (rGBM, *p53*wt), the concomitant treatment of AZD1152 and TTFields significantly increased the proportion of cells with DNA content of >4n (48.5% ± 4.1%) compared with control (28.4% ± 2.4%, *p* < 0.001) and TTFields treatment alone (36.6% ± 6.0%, *p* < 0.001), while the fraction of cells with 4n DNA content significantly decreased (45.5% ± 3.4% vs. 61.4% ± 2.3% and 53.5% ± 6.2%, *p* < 0.01). For HT16360-1 (rGBM, *p53*wt), similar significant effects were observed only with combination treatment (4n: 46.6% ± 5.5%; >4n: 45.3% ± 7.5%) compared with control (4n: 53.5% ± 4.1%, *p* < 0.001; >4n: 37.2% ± 3.6%, *p* < 0.05), but not compared with TTFields treatment alone (4n: 48.2% ± 9.0%; >4n: 40.5% ± 7.2%). HT18816 (rGBM, *p53*wt), in turn, exhibited a very stable DNA content of 2n, both with the single treatments (5 nM AZD1152: 79.0% ± 2.4%; TTFields: 77.5% ± 2.4%) and with the concomitant treatment (76.9% ± 6.1%).

## 3. Discussion

The aim of this study was to investigate the preclinical efficacy of the concomitant treatment of TTFields and Aurora B kinase inhibition by AZD1152 in glioblastoma cells. We demonstrated that in the established long-term GBM cell lines, U87-MG and U87-MG^shp53,^, and, importantly, in primary cultures from ndGBM and rGBM tissue samples, the cytotoxic effect of TTFields plus AZD1152 was significantly higher than the effect of each treatment alone.

In addition to surgery, radiation- and chemotherapy, treatment with electric fields, is increasingly incorporated into cancer therapy [25,26]. The EF-14 trial (ClinicalTrials.gov identifier NCT00916409) demonstrated that using TTFields in addition to adjuvant TMZ chemotherapy after combined chemo-radiation for ndGBM patients led to a significant prolongation of overall survival and double the rate of two-year survivors [27]. However, there is still a need to improve glioblastoma treatment, as nine out of ten patients do not survive five years past diagnosis [25].

In addition to varying the start and duration of TTFields treatment, which is being tested in ongoing clinical trials (EUDAMED-No. CIV-18-08-025247 and ClinicalTrials.gov identifier NCT03705351), identifying substances that might enhance the effects of TTFields on mitotic cells is an attractive strategy to improve treatment. Early in vitro data have shown that adding TTFields to anti-mitotic agents such as paclitaxel significantly reduces the median effective dose of the chemotherapy. Consequently, paclitaxel concomitant with TTFields was translated to clinical trials as a promising signal for safety and efficacy was seen [28] and evaluated in a phase II clinical trial for pancreatic cancer [29].

Aurora kinases are critical enzymes in the process of chromosomal segregation and cytokinesis in every cell type. Overexpression of Aurora kinases A and B is associated with malignant cell growth in various solid tissues and in myeloid cells, and can lead to acute myeloid leukemia (AML) [30,31]. Therefore, Aurora kinases are an ideal target for oncotherapy aiming at the selective inhibition of single kinase expression or function [32].

In gliomas, an overexpression of Aurora B kinase is associated with giant and multinucleated cells [17,33]. Inhibition of Aurora B kinase, in turn, results in a dysregulated connection between kinetochore and microtubule during mitosis, affecting the orientation of sister chromatids to opposite spindle poles and disrupting cytokinesis [19]. Inhibition of the CPC may lead to polyploidy, mitotic arrest, and cell death [34,35,36]. Xenograft experiments in mouse models have shown a tumor reduction after AZD1152 treatment in colon-, lung-, and hematological cancer [37]. Alafate et al. have identified Aurora B kinase as a therapeutic target in temozolomide resistant glioblastoma cells [38]. Clinical trials revealed that treatment with the selective Aurora B kinase inhibitor AZD1152 in patients with advanced solid tumors or AML was associated with an improved progression-free survival [39,40,41]. Frequent adverse events were myelotoxicity, particularly neutropenia, stomatitis, and mucositis [14,15,42]. To reduce the frequency and severity of associated toxicities, a lower effective dose of AZD1152 is desirable, which might be achieved by adding TTFields to AZD1152 treatment. Our data support the potential effectiveness of this approach; however, this finding and any impact on toxicities requires confirmation in a clinical setting.

For in vitro experiments with leukemia, colon- or lung-carcinoma cell lines, a wide range of AZD1152 concentrations (3–5300 nM) was defined as the half-maximal inhibitory concentration [34,43]. In our experiments, as we expected an augmented or, ideally, an additive or even synergistic effect, we needed to define an AZD1152 concentration high enough to render a cytotoxic effect while leaving enough cells alive to detect the potential effect of adding TTFields. For proof of principle, we started our experiments with the established cell lines U87-MG and U87-MG^shp53^, as our group has shown previously that selective Aurora B kinase inhibition leads to polyploidy and non-apoptotic cell death independent of p53 [19], while others have shown an interaction between Aurora kinase expression and p53 status [44]. For U87-MG, we determined an IC_50_ of 25 nM AZD1152, whereas U87-MG^shp53^ was less sensitive with an IC_50_ of 50 nM AZD1152. However, the combined treatment of AZD1152 and TTFields not only led to a reduction in the number of U87-MG and U87-MG^shp53^ cells but also to enhanced morphological changes, such as an increased number of cells with multiple nuclei, compared to each treatment alone.

For the treatment of primary GBM cultures, titration experiments were performed to identify the most suitable AZD1152 dose for each. IC_50_ was in the range between 14 nM and 75 nM AZD1152. However, a low effective dose of AZD1152 between 5 nM and 30 nM was selected because the use of low doses of the drug allows a better evaluation of concomitant treatment with TTFields. With regard to a possible clinical trial, based on the high efficacy of the combined treatment of TTFields with AZD1152, lower doses of Aurora B kinase inhibitor could be used, which would be associated with a reduced risk of toxicities.

Furthermore, we searched for mutations in exon 5–9 of the *TP53* gene and evaluated p53 expression in our ndGBM and rGBM cultures by western blot analysis. We identified TP53 C275F mutation, which lies within the DNA-binding domain of the TP53 protein [45]. C275F has been identified in the scientific literature [46,47,48] but has not been biochemically characterized, therefore its effect on TP53 protein function is unknown. Furthermore, we detected TP53 D208V mutation. To the best of our knowledge, there are no data about the TP53 D208V mutation, or its biochemical function, respectively. Various *p53* statuses were identified with mutations in exons 6 and 8, respectively, of the *TP53* gene in ndGBM and loss of p53 expression in two out of three rGBM cultures. The reason for that needs to be determined. In a previous study, we have shown that p53 accumulates in U87 cells [19]. This was also shown for the human colon carcinoma cell line HCT-116 by others [49]. The p53 protein decrease seems to be the result of other yet to be determined effects. One possible mechanism may be an amplification and overexpression of the negative p53 regulator MDM2, which occurs only in *p53* wild-type GBM cells and leads to increased degradation of p53 [50]. In HT16360-1 (rGBM, p53wt), no mutation was found in the analyzed hot-spot areas and p53 expression was detected, which is in line with the strong EGFR and ErbB2 expression and polyploid state of that culture, suggesting an alternative route of malignization. TP53 regulates the postmitotic checkpoint after Aurora B inhibition. It might be speculated that an increase in HT16360-1 (rGBM, *p53*wt) cell line AZD1152 concentration leads to an increase in DNA damage. Increasing the number of p53 keeps cells in checkpoint arrest for a longer period of time, allowing damages to be repaired and cells to respond to AZD1152 to a lesser extent. Regardless of p53 status, we demonstrated a pronounced cytotoxic effect in all five primary glioblastoma cultures after treatment with TTFields and a low effective dose of AZD1152, supporting our data from the analysis of U87-MG and U87-MG^shp53^ cells.

Interestingly, the primary rGBM culture HT18816 (rGBM, *p53*wt), which like all rGBM cultures had been pretreated by radiation and temozolomide-based chemotherapy, had a high proportion of diploid cells (76.5% ± 3.0%), while HT16360-1 (rGBM, *p53*wt) and HT18328-3 (rGBM, *p53*wt) had an average of only 6.4–10.1% of diploid cells. This cannot be explained by the *MGMT* methylation status. HT18816 (rGBM, *p53*wt) might be protected by its unmethylated *MGMT* promoter against the destabilizing effect of alkylating TMZ treatment, while HT16360-1 (rGBM, *p53*wt) is *MGMT*-methylated and thus polyploid. However, HT18328-3 (rGBM, *p53*wt) is *MGMT*-unmethylated and polyploid, suggesting that the observed ploidy is independent of the *MGMT*-status in those primary rGBM cultures. Treatment with either AZD1152 or TTFields, or both, led to only a minor increase in ploidy in primary cultures of rGBM compared to those of ndGBM. However, the cytotoxic effect observed in all three primary rGBM cultures was comparable with each other, suggesting that cellular instability due to polyploidy might not be the only reason for the observed cytotoxic effect.

Further studies are needed to elucidate the underlying molecular pathways associated with our observed effects of a concomitant treatment with AZD1152 and TTFields on glioma cells. Recently, our group reported that AZD1152 treatment leads to a mitotic catastrophe and cell death via a caspase-3 independent pathway, irrespective of the p53 status of the cell. Knowing that TTFields induces several modes of cell death such as immunogenic death, autophagy, necroptosis, and others (reviewed by Tanzhu et al. [51]), which are not yet fully understood, we only can assume which molecular cascades are affected by a combination treatment with a drug also affecting cytokinesis.

In view of the wide variety of genetic defects found in GBM, the number of primary cultures analyzed in our study might be too small to allow for a generalized statement.

We consider our study to be a proof of principle for the hypothesis that Aurora B kinase inhibition, together with TTFields, could be effective in glioblastoma treatment, and potentially also allow for dose-reduced concentrations of the inhibitor. However, we do not know whether the Aurora B kinase inhibitor sufficiently penetrates the blood–brain barrier, particularly in combination with TTFields, where local pharmacokinetics might be altered. This should be investigated in subsequent animal studies. In addition, further experiments using glioblastoma stem cell organoids as recently shown for the combination of a Mammalian Target of Rapamycin (mTOR) inhibitor and TTFields might validate our findings [52].

## 4. Materials and Methods

### 4.1. Cell Culture

The permanent glioma cell line U87-MG was authenticated by single nucleotide polymorphism (SNP) characterization (Multiplexion GmbH, Heidelberg, Germany). Its stable p53-deficient counterpart U87-MG^shp53^ was generated by transduction with the retroviral small hairpin (sh)RNA vector pRVH1-shp53-Hygro and has been described previously [36]. The use and further molecular analysis of primary cultures of ndGBM and rGBM from patients was approved by the local ethical committee (#EK 323122008) of the Medical Faculty Carl Gustav Carus, TU Dresden. After obtaining oral and written consent, primary glioblastoma cultures HT12347, HT18584, HT16360-1, HT18816 and HT18328-3 were prepared by using the Brain Tumor Dissociation Kit (P) (Miltenyi Biotec GmbH, Bergisch Gladbach, Germany). All cell lines were cultivated in DMEM, containing 4.5 g/L glucose supplemented with 10% *v*/*v* heat-inactivated FBS, 10 mM HEPES, 100 U/mL penicillin, and 0.1 mg/mL streptomycin (all from Life Technologies, Carlsbad, CA, USA) at 37 °C and 5% CO_2_ in a humidified incubator. U87-MG^shp53^ cells were maintained under selection with 400 mg/mL geneticin.

### 4.2. Analysis of Genomic p53 Mutations

Mutational analysis of coding regions of *p53* known to contain the DNA-binding domain and the hot spot mutation sites in exon 5–9 has been described previously [53]. Amplification of exons 5, 6, 7 and 8/9 were performed by Phusion DNA polymerase. PCR products were subsequently purified using the GeneJET PCR Purification Kit (Thermo Fisher Scientific, Waltham, MA, USA) and sequenced by Microsynth AG, Balgach, Switzerland. Sequencing data were compared with sequence of wild-type *p53* using ApE—A Plasmid Editor v2.0.47 (University of Utah, Salt Lake, UT, USA).

### 4.3. Treatment Schemes

For treatment with TTFields alone or in combination, the inovitro™ system (Novocure^®^, Root, Switzerland) was used as described previously [2]. The inovitro™ system comprises a TTFields generator and base plate containing eight ceramic dishes per plate. Next, 5 × 10^4^ glioblastoma cells were plated in triplicate on 22 mm round, poly-L-lysine coated coverslips, which were placed inside the ceramic dishes. Following overnight incubation, the dishes were filled with 2 mL medium. TTFields (1.6 V/cm RMS, 200 kHz) were applied for 72 h, where the orientation of the TTFields was switched 90° every 1 s, thus covering many of the orientation axes of cell divisions, as previously described by Kirson et al. [4]. Medium was changed every 24 h. For titration of AZD1152-HQPA (AbMole BioScience, Houston, TX, USA) or the combined treatment with TTFields, AZD1152 was added to the medium after overnight incubation of plated cells at concentrations of 5-100 nM. Treatment was applied for 72 h, with medium changes, including fresh addition of AZD1152 every 24 h.

### 4.4. Cell Viability Assays

Subsequently, inhibition of cell growth was quantitatively analyzed based on cell counting after propidium iodide (PI; Miltenyi Biotec GmbH, Bergisch Gladbach, Germany) staining, using flow cytometer EC800 (Sony Biotechnology, San Jose, CA, USA) or MACSQuant Analyzer 10 (Miltenyi Biotec).

### 4.5. Flow Cytometry

For immunophenotyping of generated primary glioblastoma cultures, cells were stained with anti-CD44-VioBlue, anti-EGFR-APC, and anti-ErbB2 (CD340)-PE (Miltenyi Biotec GmbH, Bergisch Gladbach, Germany) and with anti-GFAP-PE (Miltenyi Biotec) using the Inside Stain Kit (Miltenyi Biotec) according to the manufacturer’s instructions. Appropriate isotype controls were included in each measurement. For analysis of DNA content, treated glioblastoma cells were fixed with 70% ice-cold ethanol overnight and stained for 30 min with PI (1:50, Invitrogen, Waltham, MA, USA) in PBS containing 0.5% BSA. Stained cells were measured by MACSQuant Analyzer 10 flow cytometer (Miltenyi Biotec) and analyzed by FlowJo software version 10.6.2 (FlowJo, Vancouver, BC, Canada).

### 4.6. Western Blot Analysis

Untreated and AZD1152-treated primary glioblastoma cells were lysed in lysis buffer (10 mM Tris-HCl, pH 8.0; 140 mM NaCl; 1% Triton X-100; 1× Halt Protease and Phosphatase Inhibitor Cocktail (Thermo Fisher Scientific, Waltham, MA, USA)). Then, 50 µg protein/lane were subjected to SDS-PAGE under reducing conditions and blotted on PVDF membrane using semidry Western Blotting. After blocking the PVDF membrane with 5% BSA, p53 was detected using a monoclonal rabbit anti-human p53 (7F5) antibody (1:1000, Cell Signaling, Danvers, MA, USA), followed by HRP-conjugated anti rabbit IgG secondary antibody (1:1000; Dako, Hamburg, Germany). To demonstrate equal loading, PVDF membranes were subsequently stained using an HRP-conjugated anti-GAPDH (0411) antibody (1:200; Santa Cruz, Dallas, TX, USA). Visual capturing of proteins was performed by Luminata Forte Western HRP substrate (Merck Millipore, Burlington, MA, USA) and G:Box Chemi XX9 (Syngene, Cambridge, UK). Data were analyzed by Fiji software (ImageJ 1.53c, National Institutes of Health, Bethesda, MD, USA).

### 4.7. Light and Confocal Laser Scanning Microscopy

To detect morphological changes, treated U87-MG and U87-MG^shp53^ cells in comparison to untreated cells were fixed with 100% methanol, stained with 0.5% crystal violet (Sigma-Aldrich, St. Louis, MO, USA), and analyzed by an inverted light microscope (Nikon eclipse TS100, Nikon, Tokio, Japan).

Treated glioblastoma cells were fixed using 4% paraformaldehyde in PBS. Subsequently, cell membranes or cytoskeleton were stained with Alexa Fluor 647 conjugated Wheat Germ Agglutinin (WGA; 1:200, Life Technologies, Carlsbad, CA, USA) or anti α-tubulin antibody (1:500, Sigma-Aldrich, St. Louis, MO, USA), respectively, followed by a secondary anti-mouse IgG-FITC antibody (1:200, Jackson ImmunoResearch, Ely, UK) according to the manufacturer’s protocols. DNA was stained using Hoechst 33342 (1:50,000, Invitrogen). The coverslips were placed upside down in a drop of Vectashield mounting medium (Vector Laboratories, Burlingame, CA, USA) on a microscope slide. The images were captured by a confocal laser scanning microscope (Leica SP5, Leica, Wetzlar, Germany) and analyzed by Fiji software (ImageJ 1.53c, National Institutes of Health, USA).

### 4.8. Quantification of Multinuclear Cells

The plugin “Cell Counter” of the Fiji software (ImageJ 1.53c, National Institute of Health, USA) was used to count cell nuclei in 4–7 confocal laser-scanning-microscope pictures per preparation.

### 4.9. Statistical Analysis

Data are expressed as mean ± SD, and the statistical significance of differences was assessed using GraphPad Prism 6 (GraphPad Software, Boston, MA, USA) or IBM SPSS Statistics 25 (IBM, Armonk, NY, USA). Differences between groups were compared using the Mann–Whitney U test, and were considered significant at values of * *p* < 0.05, ** *p* < 0.01, and *** *p* < 0.001. All experiments were performed in triplicates and repeated at least two times.

## 5. Conclusions

We demonstrated that treatment with selective Aurora B kinase inhibition by AZD1152 and TTFields has a more pronounced cytotoxic effect than either treatment alone, not only in ndGBM but also in rGBM cells. This is encouraging, even just for the treatment of rGBM, as effective treatments are still a long way off.

## Figures and Tables

**Figure 1 ijms-24-05016-f001:**
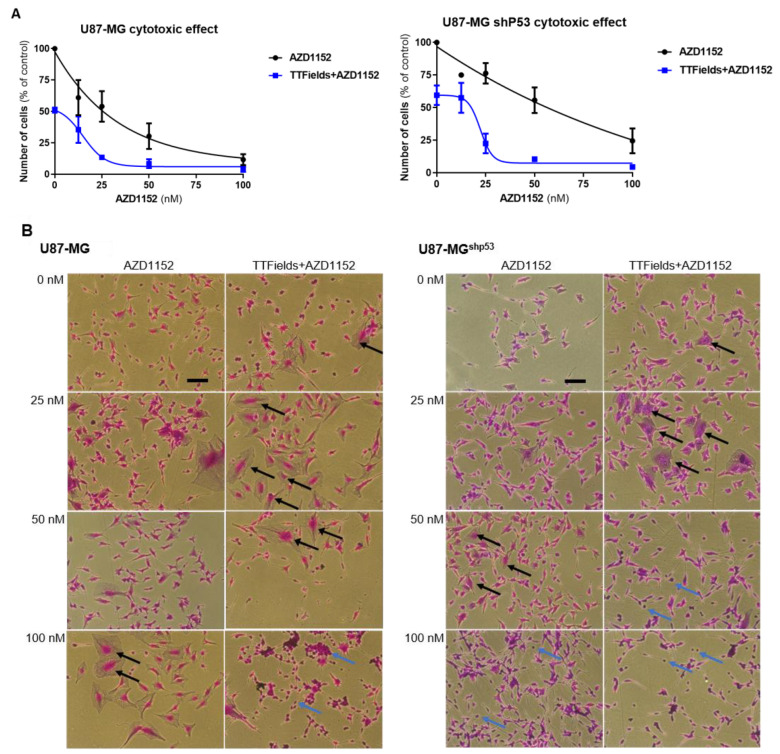
Cytotoxic effect in established cell lines. Increased efficacy of the concomitant treatment of AZD1152 and TTFields in U87-MG and U87-MG^shp53^ cells. Both glioma cell lines were treated with various AZD1152 concentrations and TTFields (200 kHz, 1.6 V/cm RMS) for 72 h. (**A**) The number of cells was determined at the end of treatment and is expressed as percentage of control. Data represent mean ± SD (N = 3 for AZD1152 and the line of TTFields + AZD is based on 2 independent experiments with 8 repeats for each concentration). For better comparison, data from both cell lines together are presented in Appendix A. (**B**) Formation of multinuclear and pyknotic cells was detected under inverted microscope after staining with crystal violet. Black arrows mark multinuclear cells and blue arrows mark pyknotic cells. The size scale in the control group corresponds to 50 µm. Enlarged views are presented in Appendix A.

**Figure 2 ijms-24-05016-f002:**
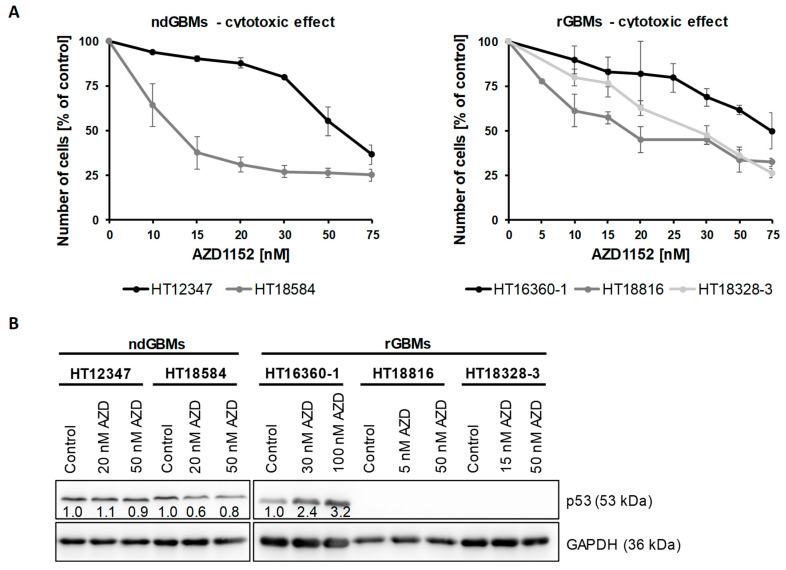
Sensitivity towards AZD1152. Treatment of primary glioblastoma cultures with increasing AZD1152 concentrations results in varying degrees of cell viability and different p53 responses. (**A**) Cells were treated with AZD1152 (5–75 nM) for 72 h and subsequently their viability was determined by flow cytometry using propidium iodide. Data represent mean ± SD (N = 9). (**B**) p53 expression was detected by immunoblotting upon treatment with different AZD1152 concentrations for 72 h, at which approximately 75% and 40% of cells survived, respectively (N = 2). Equal loading was confirmed using a GAPDH antibody. Band intensity of p53 was normalized to those of GAPDH and compared to control.

**Figure 3 ijms-24-05016-f003:**
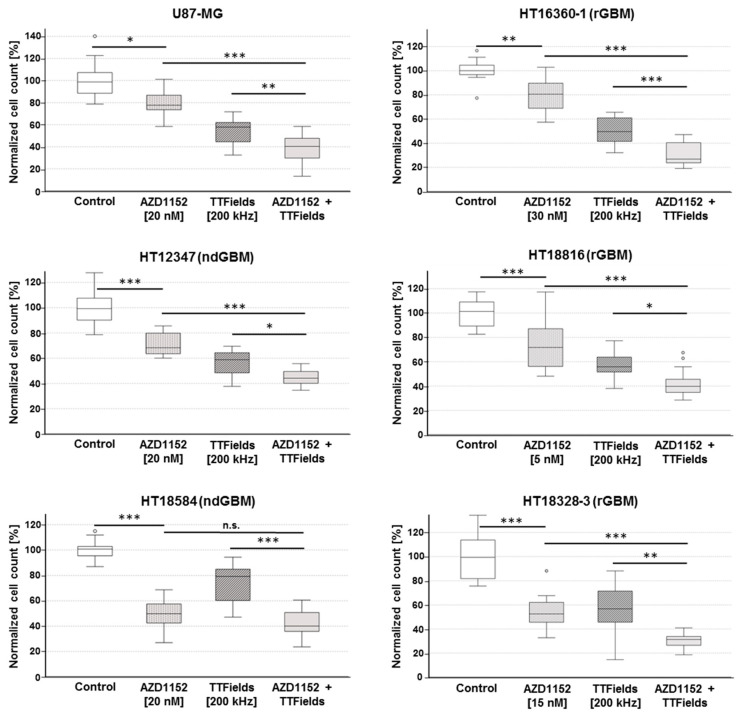
Cytotoxic effect of concomitant treatment. AZD1152 plus TTFields increase cell death. Boxplots show cell counts of U87-MG and primary glioblastoma cultures normalized to control after treatment with AZD1152 alone, TTFields alone, and the concomitant use of AZD1152 and TTFields for 72 h. Data represent median, minimum, maximum, first quartile, third quartile and ° outlier (N ≥ 12). * *p* < 0.05, ** *p* < 0.01, *** *p* < 0.001.

**Figure 4 ijms-24-05016-f004:**
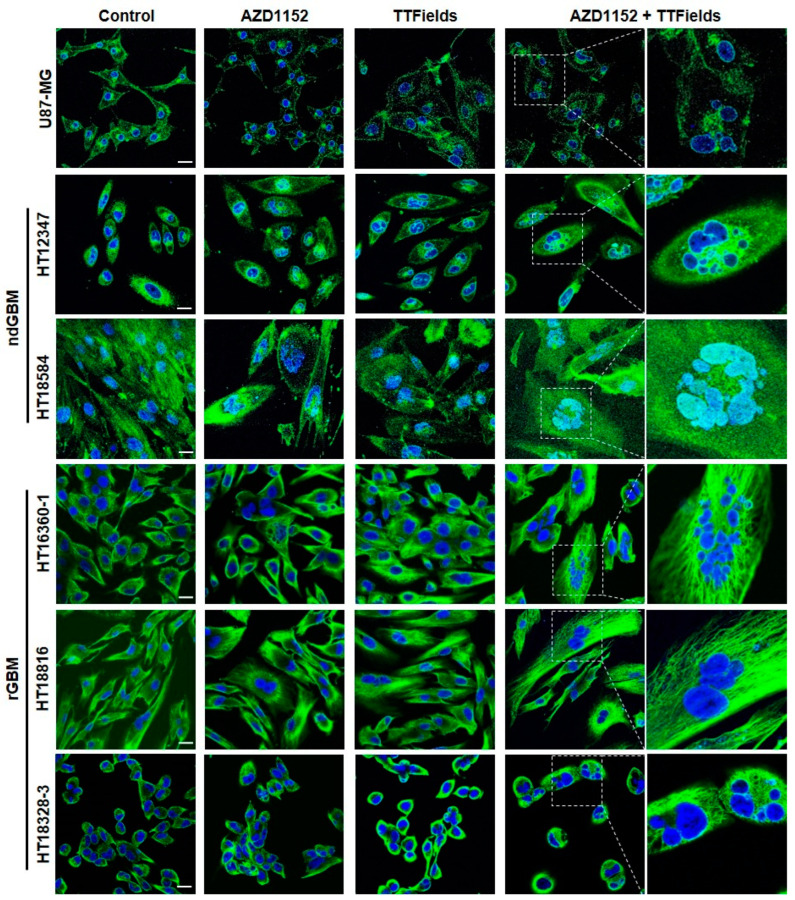
Morphological changes. Concomitant treatment of AZD1152 and TTFields enhances the morphological changes of the nucleus and cell shape in U87-MG and the primary glioblastoma cell cultures. Representative confocal laser scanning microscopic images after 72 h treatment with AZD1152 alone, TTFields alone, and AZD1152 plus TTFields are shown. Cell morphology was visualized in green with WGA (cell membrane staining, U87-MG and ndGBM) or anti-α-tubulin (cytoskeleton staining, rGBMs). DNA was stained with Hoechst 33342 (blue). The size scale in the control group corresponds to 25 µm.

**Figure 5 ijms-24-05016-f005:**
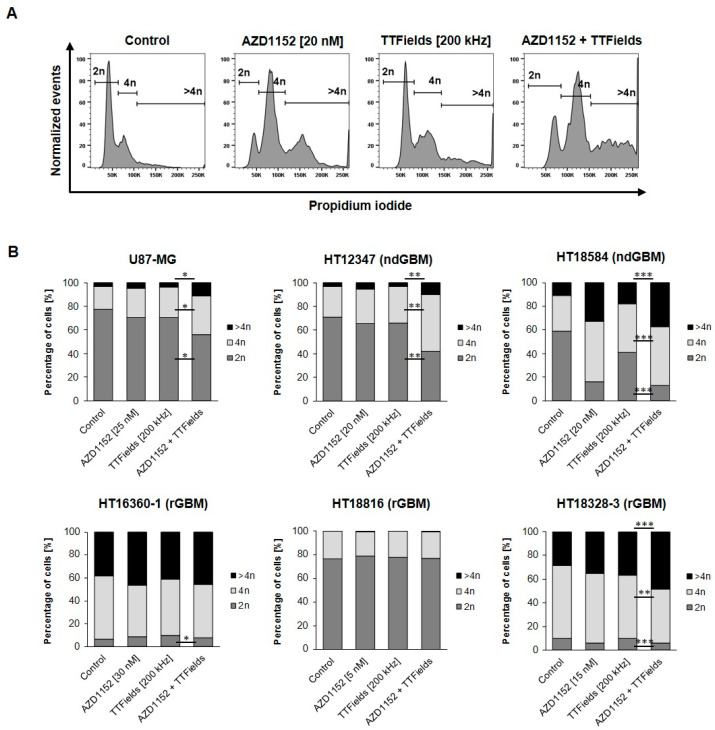
Effect on ploidy. Low concentrations of AZD1152 increase TTFields-induced polyploidy in U87-MG, ndGBMs and partially in rGBMs. Cells were treated with AZD1152 alone, TTFields alone, and AZD1152 plus TTFields for 72 h. DNA content was determined by flow cytometry using propidium iodide. (**A**) Representative flow cytometric analysis of HT18584 (ndGBM, p53mut) are shown. (**B**) Data represent mean percentage of cells with DNA content of 2n, 4n and >4n for each treatment (N ≥ 4). * *p* < 0.05, ** *p* < 0.01, *** *p* < 0.001.

**Table 1 ijms-24-05016-t001:** Characterization of cell cultures. Glioblastoma tissue pathology and immunophenotyping of established primary glioblastoma cultures. (**A**) Histological findings of the glioblastoma tissue were performed in the Department of Neuropathology of the University Hospital Carl Gustav Carus, TU Dresden. (**B**) Expression of CD44, GFAP, EGFR and ErbB2 was determined by flow cytometry from generated primary glioblastoma cultures (N = 3). Furthermore, the *p53* mutation status of these primary cultures was identified by PCR.

(A)					
Histological Findings	HT12347	HT18584	HT16360-1	HT18816	HT18328-3
Proliferation index (Ki-67)	15%	<5–40%	10–60%	<5–20%	10–45%
GFAP	++	++	+	++	+
*IDH* status	*IDH* wt	*IDH* wt	*IDH* wt	*IDH* wt	*IDH* wt
*MGMT* promotor status	Methylated	Methylated	Methylated	Unmethylated	Unmethylated
Diagnosis	ndGBM	ndGBM	rGBM	rGBM	rGBM
**(B)**					
**Immunopheno-Typing**	**HT12347**	**HT18584**	**HT16360-1**	**HT18816**	**HT18328-3**
CD44	99.0% ± 0.9%	91.2% ± 8.0%	99.1% ± 0.8%	98.8% ± 0.4%	99.3% ± 0.1%
GFAP	19.9% ± 12.0%	51.5% ± 8.9%	13.6% ± 1.8%	6.8% ± 4.4%	61.6% ± 7.2%
EGFR	78.4% ± 18.1%	33.5% ± 8.3%	73.2% ± 15.3%	20.8% ± 19.0%	56.2% ± 8.1%
ErbB2	53.7% ± 1.6%	95.9% ± 0.8%	76.3% ± 8.4%	21.9% ± 5.3%	30.6% ± 9.6%
*p53* mutation status	Point mutation in exon 8 AS 275: C → F	Point mutation in exon 6 AS 208: D → V	No mutation in exon 5–9	No mutation in exon 5–9	No mutation in exon 5–9

## Data Availability

All relevant data presented in this study are available in the current manuscript and the Appendix A. Further provisional data are provided on request from the corresponding author.

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
