# Peer review of "Aurora B Kinase Inhibition by AZD1152 Concomitant with Tumor Treating Fields Is Effective in the Treatment of Cultures from Primary and Recurrent Glioblastomas"

_ijms, 2023, doi:10.3390/ijms24055016_

Round 1

Reviewer 1 Report

This is a carefully done study and the findings are of considerable interest. A few minor revision are list below.

Suggestion:

1. In figure 3, it would be even better if Combination index (CI) were added in this good paper.

2. To discuss the signaling pathway involved in the inhibition caused by combinational therapy, the experiment could also be determined by phosphoproteomics or other bioinformatics to provide more information.

Reviewer 2 Report

In this study, Krex et al. investigated the roles of the aurora B kinase inhibitor AZD1152 and tumor treating fields (TTFields) in glioblastoma's growth and cell death (GBM) cells. This study has interesting observations; however, mechanisms under these observations are lacking. The manuscript could be improved to reach publication quality.

[1] This study mainly measured the loss of cell numbers (cell count following treatment) and cell morphology change. The observations from these measurements are superficial. Cell growth, proliferation, and the type of cell death are not defined. Effective cancer treatments directly lead to two outcomes, suppressing cell growth/proliferation and increased cell death, one or both of which should be tested.

[2] Proliferation should be measured by counting the cell number change over a period (e.g., one week) and measuring the change in proliferation markers.

[3] The type(s) of cell death (apoptosis, autophagy, necroptosis, ferroptosis) should be defined using at least two methods for each type of cell death. For example, apoptosis should be measured by TUNEL assay, western blot of cleaved caspase 3, and flow cytometry assays of annexin V and sub0G1 peaks. Furthermore, at least two inhibitors for each type of cell death and knockdown/out of pathway genes should be performed.

[4] It is better that a mechanism will be provided for the observations in this study.   

Reviewer 3 Report

The authors of the article reported interesting results regarding the approach of Tumor Treating Fields in combination with Aurora B kinase inhibitor for glioblastoma (GBM) treatment. The authors evaluated cytotoxic effects on U87-MG cells, also considering the expression of p53. They also evaluated the treatment on primary cultures newly diagnosed and recurrent glioblastoma.

I have no substantial revisions to propose. The work has been carried out very well, the results are presented exhaustively and all data have robust scientific solidity. The graphs and images are beautiful, especially those related to the confocal microscope. I would make two minor suggestions.

The first is for the description of AZD1152. I suggest describing the inhibition mechanisms in more detail. Furthermore, in the discussions, I suggest mentioning, as a research perspective, the evaluation of this combined treatment on GBM stem cells in addition to preclinical in vivo models.

Reviewer 4 Report

The article by Krex and coworkers explores the cytotoxic effect of the combination of Tumor Treating Fields (TTFields) and the AZD1152 antimitotic drug treatment on glioblastoma (GBM) cells. The authors tested both GBM cell lines and patient-derived GBM cells, derived from both newly diagnosed (ndGBM) and recurrent glioblastoma (rGBM). The work is well-written and potentially interesting for its therapeutic implication for GBM treatment, but, in my opinion, the GBM primary cultures analyzed (5 in total and also divided into two subgroups of 2 and 3 samples, respectively) are too few and too heterogeneous to obtain strong results. The authors themselves acknowledge this objective limit in their work at the end of the discussion. Due to the small number of samples and the extreme heterogeneity, it is not possible to assess if there is any correlation between the response obtained and the different features analyzed, a correlation that would deserve to be further investigated in order to identify the cases in which the different therapies could be truly effective. Moreover, the work is only descriptive, not investigating the regulatory mechanism behind the interesting findings observed, such as the loss of p53 expression in some rGBM samples, although the absence of p53 mutations (the authors merely speculate on MDM2 involvement). As it is, unfortunately, I cannot recommend publication in IJMS.  I would suggest increasing the number of samples analyzed, and improving the statistical analysis to obtain reliable results.

Round 2

Reviewer 2 Report

The authors provided acceptable reasons for why they would not perform the experiments I suggested.

Reviewer 3 Report

The quality of the work was high in the first version. The authors have integrated my minor comments, therefore, I can confirm acceptance for publication.

Reviewer 4 Report

I have carefully read the response of the Authors as well as the revised version of the paper. In the first round, I opted for rejection (also in consideration of the optimum journal that IJMS is) due to the extreme heterogeneity of the response, the small number of cases analyzed, and the tendency to define the effect measured irrespective of the GBM features analyzed, such as p53 status.  In my opinion, as they are, data are not sufficient to demonstrate/suggest this.

I totally agree with the Authors that increasing the number of samples analyzed and/or investigating the mechanisms involved requires more time than the ones usually given for revision. In the absence of new data, unfortunately, the article can be a mere, even if valuable, proof of concepts, on the higher cytotoxic effect of the combinatory treatment, as the authors themselves affirm. Please, this has to be clearly highlighted in the article, mainly in the Abstract and in the Discussion. Moreover, I recommend improving data analysis, presentation, and discussion, especially relative to the heterogeneity in the cytotoxic effect obtained with respect to the p53 status of the GBM samples. 

Below are the specific comments:

1)     Comparison between U87-MG cells and U87-MGshp53 (figure 1A and 1B) and relevance of p53.

-        The experiment in Figure 1A should be done in triplicate, in consideration also of some high standard deviations obtained.

-        It would be valuable to add a graphic in which the effect of the same treatment on the two different U87-MG and U87-MGshp53 cell lines is directly compared to highlight if and how p53 absence might affect the treatment.

-        Lines 117-118: authors report “treatment with AZD1152 alone with a half-maximal inhibitory concentration (IC50) around 20 nM (Fig. 1A)”. Could it be closer to 25 nM, that is the concentration present in the graph in Fig. 1A?  Please change and/or explain.

-        Taking into account all the data, it seems to me that response to AZD1152 is affected by p53 absence, because in the U87-MGshp53 cell line lacking p53 both AZD1152 single and combined effects are attenuated. Please comment on this in the text.

-        Some pictures inside Figure 1B are not clear to me (i.e. in the image of U87MG cells after AZD1152 25nM treatment there are cells that might seem multinucleated, but are not indicated by arrow). I would recommend increasing magnification to clearly indicate multinuclear and pyknotic cells and highlight differences between the two treatments which is the scope of the experiment.

-        I would really appreciate a quantification of the multinuclear and pyknotic cells (in part reported in Figure s1). 

-        A sentence on this experiment might be eventually added in the Abstract.

2)     Analysis of Primary glioblastoma cultures

-        I would modify the sentence in lines 190-192 ”These showed variable sensitivity to AZD1152 treatment regardless of their  p53 status or whether they were established from ndGBM or rGBM samples. (Figure 2A)”. I would suggest giving less emphasis to this concept, due to the small cases analyzed, and also because the experiments might give some indications of a possible relation that has to be confirmed by larger and more detailed studies (see below).

-        It would be really valuable for the readers when authors describe the behavior of the different samples to add after the GBM label info on ndGBM/rGBM and p53mut/wt status between parenthesis, i.e. lines 190-193: While HT18584 (ndGBM, p53mut) and HT18816 (rGBM, p53wt) exhibited high sensitivity with IC50 around 14 nM and 18 nM…..

Please check and add this info throughout the text.

-        How do the p53 mutations found in the ndGBM affect p53 protein, do they alter stability or functionality or is it unknown? Could the different ndGBM behavior be due to the different p53 mutations found?

-        The results of Western blot are quite interesting and perhaps should be better analyzed and/or discussed, to highlight possible indications and suggestions. Below are some considerations, that could be evidenced/commented on in the text.

1. There is no clear relation between the presence of the p53 mutations analyzed by PCR and the p53 protein amount identified by Western blots. Is it expected?

2. Is AZD1152 expected to cause p53 protein decrease in GBM or tumor cells in general? What does occur in U87-MG cells? In the case of HT12347 ndGBM (in which the cytotoxic effect is obtained at higher doses), a clear p53 downregulation is not evident whereas it is clearer in the HT18584 sample, which is the one that responds more to AZD1152.

3. At the same time, the HT16360-1 rGBM sample which shows a significant increase in p53 protein amount (could you suggest how?) is also the one that responds less to AZD1152.

According to me, it is not possible to exclude the existence of a correlation between p53 and the cytotoxic effect obtained, as already highlighted above. Moreover, also experiments on U87 with or without p53 seem to go in the same direction (see previous comment).

3)     Cytotoxic effect of combined treatment (figure 3).

-        It would be interesting to add also a graphic relative to U87-MGshp53

-     Please could you also add the significance of the difference between control and AZD1152 treatment alone?

-        Line 220-221: (HT18584: 49.8% with 20 nM AZD1152; HT18328-3: 52.5% with 15 nM AZD1152). Do these values agree with the one reported in Figure 2A? Please check.

-        Even though the combination of the treatment is always more effective than the treatments alone, it seems to me that the total amount of final living cells and also the amelioration in the percentage of efficacy is variable between the different samples and needs to be better studied. It is not possible to exclude that this difference depends on p53 status (mutation, presence/absence, downregulation/upregulation) or other GBM characteristics mentioned in the article. I would suggest revising the sentence at lines 231-232 and introducing a small comment on this.

4)     Morphological changes (figure 4).

-        This Figure is really beautiful! I would just suggest adding some magnifications also of the AZD1152 and TTF treatment alone to make differences more evident, that is the point of the article.

Please check that reference to Supplementary Figure S1 is present in the text and also that quantification is described in the Methods. 

Line 393-395. I think that the previous version of the sentence is definitely more adequate.

Round 3

Reviewer 4 Report

The authors have addressed most of my comments. I really appreciate the Supplementary Figures added and the answers they gave to my comments. I would just suggest inserting some of these clarifying comments also in the manuscript and checking that Supplementary Figures are cited in the text. Probably the order of the Supplementary Figures should be changed, according to their appearance in the Results.

Specifically, please consider incorporating in the manuscript the following sentences present in the answers in the text:

“Although there is a difference between U87 and U87-MGshp53 cell lines we performed a test of significance but could not substantiate a statistically significant difference neither for the single treatment with AZD1152 nor in combination with TTFields”.

“TP53 C275F lies within the DNA-binding domain of the Tp53 protein (PMID:21760703). C275F has been identified in the scientific literature (PMID: 25847421; PMID: 26873401; PMID: 27095739; PMID 29026176) but has not been biochemically characterized and therefore, its effect on Tp53 protein function is unknown. To the best of our knowledge, there are no data about the TP53 D208V mutation, or its biochemically function, respectively”

"In a previous study we have shown that p53 accumulates in U87 cells (doi: 10.1093/carcin/bgw083). This was also shown for the human colon carcinoma cell line HCT-116 by others (DOI 10.1091/mbc.e08-08-0885 PMID 19225156). The p53 protein decrease seems to be the result of other yet to be determined effects".

"P53 regulates the postmitotic checkpoint after Aurora B inhibition. It might be speculated that in HT16360-1 (rGBM, p53wt) cell line increasing AZD1152 concentration leads to increasing DNA damage. Increasing p53 amounts keep cells for a longer time in checkpoint arrest so that damages might be repaired and therefore cells react to AZD1152 to a lesser extent"
